# In Vitro Biofilm Formation by *Malassezia pachydermatis* Isolates and Its Susceptibility to Azole Antifungals

**DOI:** 10.3390/jof8111209

**Published:** 2022-11-15

**Authors:** Eva Čonková, Martina Proškovcová, Peter Váczi, Zuzana Malinovská

**Affiliations:** Department of Pharmacology and Toxicology, University of Veterinary Medicine and Pharmacy, Komenského 73, 041 81 Košice, Slovakia

**Keywords:** *Malassezia pachydermatis*, biofilm, antifungal susceptibility, azoles

## Abstract

The yeast *Malassezia pachydermatis*, an opportunistic pathogen that inhabits the skin of various domestic and wild animals, is capable of producing a biofilm that plays an important role in antifungal resistance. The aim of this research study was to find the intensity of biofilm production by *M. pachydermatis* strains isolated from the ear canal of healthy dogs, and to determine the susceptibility of planktonic, adhered and biofilm-forming cells to three azole antifungals—itraco-nazole, voriconazole and posaconazole—that are most commonly used to treat *Malassezia* infections. Out of 52 isolates, 43 *M. pachydermatis* strains (82.7%) were biofilm producers with varying levels of intensity. For planktonic cells, the minimum inhibitory concentration (MIC) range was 0.125–2 µg/mL for itraconazole, 0.03–1 µg/mL for voriconazole and 0.03–0.25 µg/mL for posaconazole. Only two isolates (4.7%) were resistant to itraconazole, one strain (2.3%) to voriconazole and none to posaconazole. For adhered cells and the mature biofilm, the following MIC ranges were found: 0.25–16 µg/mL and 4–16 µg/mL for itraconazole, 0.125–8 µg/mL and 0.25–26 µg/mL for voriconazole, and 0.03–4 µg/mL and 0.25–16 µg/mL for posaconazole, respectively. The least resistance for adhered cells was observed for posaconazole (55.8%), followed by voriconazole (62.8%) and itraconazole (88.4%). The mature biofilm of *M. pachydermatis* showed 100% resistance to itraconazole, 95.3% to posaconazole and 83.7% to voriconazole. The results of this study show that higher concentrations of commonly used antifungal agents are needed to control infections caused by biofilm-forming strains of *M. pachydermatis*.

## 1. Introduction

*Malassezia pachydermatis*, a natural commensal of dogs’ skin, is one of the most common yeasts involved in otitis externa and dermatitis [1,2]. The pathogenicity of *M. pachydermatis* is influenced by several predisposing factors, such as dog breed, the microenvironment (increased temperature and humidity), the presence of skin folds, changes in skin pH, increased sebum production, primary bacterial infection, endocrine system disorders (hypothyroidism and diabetes mellitus) and immunosuppression [3,4]. In a study by Cafarchia et al. [5], *Malassezia* yeasts were isolated from 57.3% of dogs with otitis externa and 28.0% of dogs without otitis externa. Similar to these authors, our previous study reported a higher prevalence of the *M. pachydermatis* yeast in dogs suffering from otitis externa with pendulous ears (51.4%) compared to 34.8% in breeds with erected ears. Regarding the type of haircoat, in dogs with dermatitis, the prevalence of *M. pachydermatis* was significantly higher in both long-haired and short-haired dogs (51.5% and 45.9%, respectively) than in smooth-haired dogs (21.4%) [6]. As for the epidemiology of *Malassezia* infections, there is also a difference in the population size of *Malassezia* recovered from animals with otitis (66.0 CFU/dog) compared to healthy animals, which can play an important role in the pathogenesis of otitis externa [5]. The conversion of the commensal to the pathogenic form of *M. pachydermatis* is determined by several virulence factors, such as biofilm formation, protease, phospholipase, haemolytic factor and melanin production, and adhesion to epi-thelial cells [7,8,9]. In the chronic form of otitis externa, an increased production of viscous, dark-brown-to-black cerumen containing bacteria, yeast, exudate and debris is found in the external ear canal, which may represent a real biofilm [10].

Antifungal agents, mainly from the group of polyenes (nystatin) and azoles (e.g., clo-trimazole, enilconazole, miconazole, and posaconazole), are used to treat *Malassezia* infections [11,12]. Some authors confirm the ability of *M. pachydermatis* to form a biofilm in vitro and, at the same time, point to an increased development of resistance in biofilm-forming strains to antifungals [13]. The aim of this study was to determine the intensity of biofilm production by *M. pachydermatis* isolates obtained from healthy dogs and determine the susceptibility of cells adhered to the well surface of the microtiter plate and mature biofilms to selected antifungals.

## 2. Materials and Methods

### 2.1. Isolates Tested

The isolates of *M. pachydermatis* were obtained in cooperation with the Small Animal Clinic at the University of Veterinary Medicine and Pharmacy in Košice and a grooming salon by swabbing the external ear canal of healthy dogs of different breeds, sexes and ages.

Out of a total 165 taken samples, *M. pachydermatis* was detected by phenotypic diagnostics (macroscopic and microscopic characteristics, ability to grow with/without lipid supplementation) in 52 samples. All phenotypically positive samples recognised as *M. pachydermatis* were investigated by PCR-RFLP [14]. The internal transcribed spacer 2 region (ITS2) was amplified by PCR using the ITS3 (5′-GCATCGATGAAGAACGCAGC-3′) and ITS4 (5′-TCCTCCGCTTATTGATATGC-3′) primers (Life Technologies, Carlsbad, CA, USA) [15]. Three endonucleases, AluI, BanI and MspA1I (New England Biolabs, Ipswich, MA, USA), were used for the digestion of the PCR products and for genotype identification of *M. pachydermatis*.

Only biofilm-producing isolates (43 samples) were included for further experiment. Until the beginning of the experiment, the isolates were stored at −80 °C in a freezing medium (100 µL of 60% glycerol and 300 µL of medium—glucose, 4 g; tryptophan, 1 g; and yeast extract, 0.5 g per 100 mL).

The assay was also performed on a reference strain of *M. pachydermatis* CBS 1879 (Centraalbureau voor Schimmelcultures, Utrecht, The Netherlands).

### 2.2. Determination of Biofilm Production

The methods described by Jin et al. [16] and Bumroongthai et al. [17] were used to determine biofilm production in tested *Malassezia pachydermatis* strains with some modifications. Briefly, SAOT (Sabouraud dextrose agar supplemented with glycerol—2 mL, Tween 80—2 mL, Tween 40—5 mL, and olive oil—5 mL per litre) was used as a growth medium for *Malassezia* strains cultivated at 35 °C for 72 h. A loopful (1 µL) of strain colonies cultured on SAOT was transferred into 20 mL of SBOT (Sabouraud dextrose broth supplemented with the same ingredients as SAOT) in an Erlenmeyer flask and incubated for 72 h at 35 °C on an orbital shaker at 80 rpm. The yeasts grown in SBOT were then centrifuged and washed twice with 5 mL of phosphate-buffered solution (PBS, pH 7.2). The inoculum suspension of approximately 1–5 × 10^6^ CFU/mL of each tested strain was prepared by adjusting to a 0.1 optical density at 600 nm using a spectrophotometer (Thermo Spectronic Helios Gamma, Thermo Fisher Scientific, Leicestershire, UK).

The assay was performed in sterile, 96-well, flat-bottomed microtiter plates (Brand GMBH, Wertheim, Germany). A total of 150 µL of each isolate suspension was added into the wells of the microtiter plates in triplicate. The microtiter plates were incubated for 24 h at 35 °C on an orbital shaker at 80 rpm to allow the yeast to adhere to the surface of the plate wells. After the adhesion phase, the cell suspension was removed and the wells were washed with 150 µL of PBS. Then, 200 µL of SBOT was added into each well and the plates were incubated at 35 °C on an orbital shaker at 80 rpm for 72 h.

Afterwards, the biofilm-coated wells were rinsed twice with 200 µL of PBS and dried at room temperature for 45 min. The wells were then stained with 0.5% crystal violet solution for 45 min and washed four times with 350 µL of sterile distilled water and de-stained with 200 µL of 95% ethanol for 45 min. After destaining, 100 µL of solution was transferred into a new microtiter plate. The biofilm biomass was quantified as the amount of the crystal violet in destaining solution by measuring the OD (optical density) value at 650 nm with an ELISA microplate reader (Dynex, Prague, Czech Republic). The intensity of biofilm formation was evaluated according to Ruchi et al. [18] (Table 1). The cut-off value (ODc) was established as three standard deviations (SDs) above the mean OD of the negative control: ODc = average OD of negative control + 3 × SD of negative control. The final OD value of a tested strain was expressed as the average OD value of the strain reduced by the ODc value (OD = average OD of a strain − ODc).

### 2.3. In Vitro Susceptibility Testing of M. pachydermatis Planktonic Cells

The assay was performed using the broth microdilution standard methods M27-A3 [19], with some modifications. SBOT as the growth medium was used and the final yeast inoculum was 10^4^ CFU/mL. At first, 100 µL of SBOT was added into columns 2–12 and 200 µL of the test antimycotic at a concentration of 32 µg/mL was added in column 1. Then, 100 μL of antifungal solution was taken from well number 1, added into well number 2 and mixed well. The process was repeated up to well number 10, from which the rest (100 μL) was discarded. Using this two-fold dilution, the concentrations of antifungals in a range of 32 to 0.0625 µg/mL were prepared. The efficacy of three antimycotics, itraconazole (ITR), voriconazole (VOR) and posaconazole (POS), was tested. A total of 200 µL of SBOT was applied to column 11, which served as the negative control. After adding 100 mL of suspension of *Malassezia* isolates to the wells 1–10 and 12, the concentrations of antifungals were halved so that final concentrations of tested antifungals in the range of 16 µg/mL to 0.0313 µg/mL were obtained. Column 12, considered a positive control, contained 100 µL of SBOT and 100 µL of inoculum. The microplates were incubated at 35 °C for 72 h and then the results of the minimum inhibitory concentration (MIC) were read. The MIC was defined as the lowest concentration that inhibited 50% of yeast growth as compared to the control [19]. For a better evaluation of the MIC end points, the colorimetric reading of results was used by adding 10 µL of 0.1% resazurin (sterilised through a 0.22 µm filter before use) into each well of the microplate six hours before reading the results. Growth inhibition was indicated at an MIC that prevented the change from blue (no yeast growth) to pink (yeast growth). Finally, the absorbance for all wells in the plate was determined using the ELISA microplate reader (Dynaread, Dynex, Prague, Czech Republic) at 650 nm [20,21].

Based on the measured absorbance value, the percentage of yeast growth inhibition was calculated according to the formula:$$I\;\left(\%\right)=\frac{\left(ODpc-ODnc\right)-\left(ODs-ODpc\right)\;}{\left(ODpc-ODnc\right)}\times 100$$

*I*—percentage of growth inhibition;

*OD_pc_*—optical density of positive control;

*OD_nc_*—optical density of negative control;

*OD_s_*—optical density of sample.

The following criteria were used to classify the *M. pachydermatis* tested strains’ susceptibility: susceptible (S)—MIC sample ≤ MIC50; susceptible dose-dependent (S-DD)—MIC50 ˂ MIC sample ≤ MIC 90; and resistant (R)—MIC sample > MIC90 [13].

In addition, an epidemiological cut-off value (ECV) was determined, which is defined as the MIC threshold value according to which wild-type (WT) strains (isolates without mutation or acquired resistance mechanisms) can be distinguished from non-WT (isolates with mutation or acquired resistance mechanisms). The ECV represents, usually, an MIC that is approximately two dilutions above the modal MIC and encompasses (MIC ≤ ECV) about 95% of the results in the WT MIC distribution [22,23].

### 2.4. In Vitro Susceptibility of Adhered M. pachydermatis Cells

The cell preparation was the same as described in the Determination of Biofilm Production section. A total of 150 µL of suspension of each strain tested containing 10^6^ CFU/mL was transferred into rows A–H of wells 1–12 of the 96-well flat-bottom microplates, with the exception of well 11 (negative control), and was allowed to adhere to the microplate well surface for 24 h at 35 °C on an orbital shaker at 80 rpm. The microplate wells were then rinsed twice with 200 µL of PBS to remove nonadhered cells. Thereafter, 100 µL of SBOT was added into each well of the microplate. Subsequently, 100 µL of antimycotics was tested, prepared by binary dilution and transferred at a descending concentration in the range of 32 µg/mL to 0.0625 µg/mL into wells 1–10. The final concentration in the wells then reached from 16 µg/mL to 0.0313 µg/mL. In well number 12 (positive control), 100 µL of SBOT was added in order to keep the volume in each well the same. The plates were incubated at 35 °C on an orbital shaker at 80 rpm. After 72 h, the results (MIC) were read by employing the colorimetric method (described above) and the Elisa reader. The MIC end points were determined in microplate wells in which no biofilm bio-mass was formed (i.e., at the concentration showing more than a 95% reduction of absorbance).

### 2.5. In Vitro Susceptibility Testing of Mature Biofilm of M. pachydermatis

To obtain a mature biofilm, 200 μL of SBOT was added to the adhered cells and incubated at 35 °C by shaking at 80 rpm. After 72 h of incubation, the plates were washed twice with PBS. Subsequently, 100 μL of SBOT and 100 μL of antimycotics were added to the microplate wells in the same manner as in the susceptibility testing of adhered cells. This was followed by incubation on an orbital shaker, and the results (MIC) were read using the colorimetric method (OD 650 nm) and ELISA reader after 72 h. The MICs were defined as the lowest concentration of drug that disintegrated more than 95% of the biofilm biomass.

### 2.6. Statistical Analysis

Each experiment was repeated twice and average values were taken. The data are presented as average means (x), standard deviations (SD), mode and median. A one-way ANOVA followed by Tukey´s test was used to analyse the MIC means of different antifungal agents for planktonic cells, adhered cells or mature biofilms (GraphPad Prism 8.0.1, San Diego, CA, USA). The level of statistical significance was set as *p* < 0.05.

## 3. Results

Out of 52 *M. pachydermatis* isolates (Table 2), in 9 strains (17.3%) there was no observed biofilm formation. Weak and strong biofilm production was shown in 18 isolates (34.6%), whereas moderate biofilm production was found in 7 isolates and also in the re-ference strain *M. pachydermatis* CBS 1879. All biofilm producers (43 isolates/82.7% and re- ference strain) were included in the following assay.

Table 3 summarises the MIC data for tested azoles and ECVs (95%). For the 43 *M. pachydermatis* planktonic cells (PC), the MIC range was 0.125–2 µg/mL for itraconazole, 0.03–1 µg/mL for voriconazole and 0.03–0.25 µg/mL for posaconazole. The following MIC ranges were found for the *M. pachydermatis* CBS 1879 reference strain: 0.5 µg/mL for itraconazole, 0.125–0.25 µg/mL for voriconazole and 0.125 µg/mL for posaconazole. Significantly higher MIC ranges (*p* ˂ 0.05) were observed for adhered cells (AC; 0.25–≤16 µg/mL for itraconazole, 0.125–8 µg/mL for voriconazole and 0.03–4 µg/mL for posaconazole) and mature biofilms (MB; 4–≤16 µg/mL for itraconazole, 0.25–≤16 µg/mL for voriconazole and 0.25–≤16 µg/mL for posaconazole). No statistically significant difference was found between the MIC for voriconazole and posaconazole in planktonic and adhered cells and in mature biofilms.

The ECV of 95% was 0.5 µg/mL for itraconazole and voriconazole, and 0.25 µg/mL for posaconazole. When comparing the ECV value with MIC, two isolates of planktonic cells (4.7%) achieved an MIC > ECV for itraconazole and one isolate (2.3%) for voriconazole. A total of 38 adhered cells (88.4%) of *M. pachydermatis* strains showed an MIC > ECV for itraconazole, 27 strains (62.8%) for voriconazole and 24 strains (55.8%) for posaconazole. All of the mature biofilm strains reached an MIC > ECV for itraconazole (43/100%). An MIC > ECV was found in 36 isolates (83.7%) for voriconazole and in 41 strains (95.3%) for posaconazole in mature biofilms. In the *M. pachydermatis* CBS 1879 reference strain, the MIC > ECV was attained 100% in adhered cells and mature biofilm for itraconazole and in mature biofilms for voriconazole and posaconazole.

The susceptibility of planktonic cells, adhered cells and mature biofilms of tested *M. pachydermatis* isolates is depicted in Table 4 and Figure 1. The highest susceptibility of planktonic cells was detected for voriconazole (76.8%), followed by itraconazole (55.8%) and posaconazole (51.1%). For planktonic cells, 17 isolates (39.5%) were susceptible dose-dependently to itraconazole, 11 strains (25.6%) for voriconazole and 21 (48.9%) for posaconazole. A decreased susceptibility was observed in adhered cells, with only two strains being susceptible (4.7%) and three isolates (6.9%) being susceptible dose-dependently to itraconazole. Further, 5 strains (11.6%) of adhered cells were susceptible and 11 strains (25.6%) were susceptible dose-dependently to voriconazole. For posaconazole, for adhered cells, 4 isolates (9.3%) showed susceptibility and 15 strains (34.9%) were susceptible dose-dependently. All *M. pachydermatis* strains forming mature biofilms were resistant to itraconazole (100%). Out of 43 yeast isolates forming mature biofilms, 41 strains (95.3%) were resistant to posaconazole and 36 strains (83.7%) to voriconazole.

## 4. Discussion

A biofilm is defined as a differentiated microorganism community, consisting of a single microbial agent or of a mixture of fungal and/or bacterial species which adhere to biotic or abiotic surfaces, and which is difficult to remove [13]. The ability of *M. pachydermatis* to form biofilms was confirmed by some authors [7,9,24]. The formation of a biofilm comprises four different phases, i.e., adhesion, proliferation, maturation and dispersion. The adherence of yeast cells to the surface as well as to each other is an important step for forming a basal layer of biofilm. Following the adherence phase, the cells proliferate and form an anchoring layer that provides primary stability to the biofilm [25]. Based on the findings thus far, there is an assumption that *M. pachydermatis* can produce biofilms on the skin and in the ears of dogs. *Malassezia otitis* is manifested on cytology by an active overgrowth of the yeasts, which remain “entrapped” in the cerumen and debris and, in some cases, can be the sites of the actual biofilm formation [26]. In our study, 43 strains of *M. pachydermatis* (86.7%) were able to form a biofilm in various levels of intensity production. These results are in-line those found by Figueredo et al. [7], who reported biofilm production by 52 (95.2%) M. *pachydermatis* strains isolated from dogs with or without skin lesions. Gagana et al. [9] showed no statistical difference in biofilm formation by *M. pa-chydermatis* isolates from dogs with otitis or dermatitis and healthy dogs. Yeasts, regardless of whether they were able to cause infection or remained commensal, had the potential to form biofilms at various levels.

In an in vitro system, the maturation of a *M. pachydermatis* biofilm usually takes 72–96 h [17,27]. The architecture of a mature biofilm consists of blastoconidia organised in multilayers, a variable amount of an extracellular matrix, and water channels within the structure [8]. Although the early phase of biofilm formation lacks the extracellular matrix, its presence in the mature biofilm protects the biofilm from any physical perturbations and provides resistance against various xenobiotics. The development of infection is supported by the dispersion phase, characterised by the release and dispersion of yeast cells to new sites. The dispersed yeast cells differ from planktonic cells due to their increased virulence and adhesive features that allow them to form new biofilms [25]. Some research articles have focused on the susceptibility determination of planktonic and biofilm-forming cells to antifungals. Azole antifungals are mostly used to treat *Malassezia* infections. By comparing the susceptibility of planktonic and sessile forms in the biofilm of *M. pa-chydermatis*, Bumroongthai et al. [17] found the effectiveness of ketoconazole and itraconazole (˂0.3 µg/mL) for planktonic cells, but also found resistance to biofilm-forming cells (>16 µg/mL). Additionally, in the study by Jerzsele et al. [27], ketoconazole and itraconazole were more effective on planktonic than biofilm-forming cells.

Figueredo et al. [13] tested the susceptibility of both planktonic and sessile forms of *M. pachydermatis* to six antifungals (ketoconazole, itraconazole, posaconazole, voriconazole, fluconazole and terbinafine). They report that the MICs were significantly higher for sessile cells than planktonic cells. Up to 98.3% of sessile cells were resistant to ketoconazole, 95% to itraconazole, 93.3% to posaconazole and 90% to fluconazole and voriconazole. Their study proved that biofilm formation is responsible for antifungal resistance, and influenced by some factors such as the density of the biofilm population and the presence of an extracellular matrix, which plays an important role.

In addition to determining the effect of selected azoles against planktonic and biofilm-forming strains of *M. pachydermatis*, this study also includes the determination of the antifungal susceptibility of adhered cells, which has been not reported in investigations thus far. Although the available articles compare the effectiveness of commonly used antifungals against planktonic cells and mature biofilms, and report a higher resistance of the mature biofilm, there is no mention of whether the adhered yeast cells develop the same resistance as mature biofilms. Therefore, one of the objectives in this study was to determine whether the adhered cells can show the same resistance to the used antifungals as a mature biofilm.

Adherence is the first step of a microorganism’s colonisation of any site in the host cells. In vitro studies have shown that *M. pachydermatis* adheres to canine corneocytes in a dose- and time-dependent manner by binding proteins or glycoproteins expressed on their surface to carbohydrate ligands on canine corneocytes [3,28]. *Malassezia* cells adhering to keratinocytes have the potential to modulate the expression of a number of cytokines, chemokines and antimicrobial peptides, resulting in immunostimulation that may occur in diseases characterised by the development of skin inflammation [29,30]. When comparing the MIC results, all the antifungal agents tested were more effective against the planktonic cells than the adhered cells or mature biofilms. The highest resistance of adhered cells was shown for itraconazole (88.4%), followed by voriconazole (62.8%) and posaconazole (55.8%). In general, the best efficacy was observed for voriconazole despite the fact that the resistance of *M. pachydermatis* strains forming mature biofilms reached 83.7% (36 strains).

The results of this study indicate that the increased resistance of *M. pachydermatis* strains develops not only in mature biofilms, but also in adhered cells. At the same time, the results support the hypothesis that the treatment of *Malassezia* biofilm infections requires higher drug concentrations than those currently used. Therefore, in the chronic form of *Malassezia* infections, it is necessary to find out the susceptibility of the isolated yeast to the antifungal agents, which will be used for the treatment.

## Figures and Tables

**Figure 1 jof-08-01209-f001:**
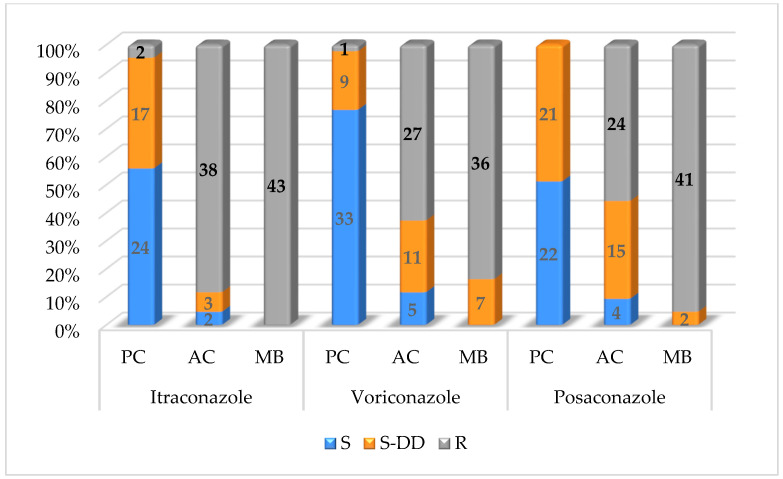
Susceptibility of planktonic cells (PC), adhered cells (AC) and mature biofilm (MB) of *M. pachydermatis* strains; S—susceptible, S-DD—susceptible dose-dependent, R—resistant.

**Table 1 jof-08-01209-t001:** The criteria for evaluation of biofilm producer.

OD Value	Biofilm Producer
OD < ODc	Non
ODc < OD < 2 × ODc	Weak
2 × ODc < OD < 4 × ODc	Moderate
OD > 4 × ODc	Strong

OD—optical density of the sample, ODc—average of the negative control + 3 × SD of negative control.

**Table 2 jof-08-01209-t002:** Biofilm formation by *M. pachydermatis* strains.

Intensity of Biofilm Production	Number of Isolates*n*/%
Non	9/17.3
Weak	18/34.6
Moderate	7/13.5
Strong	18/34.6
Total	52/100

*n*—number of isolates.

**Table 3 jof-08-01209-t003:** Evaluation of MICs (µg/mL) in tested antimycotics.

Parameter	Itraconazole			Voriconazole			Posaconazole		
	PC	AC	MB	PC	AC	MB	PC	AC	MB
Range	0.125–2	0.25–≥16	4–≥16	0.03–1 ^a^	0.125–8 ^b^	0.25–≥16 ^c^	0.03–0.25 ^a^	0.03–4 ^b^	0.25–≥16 ^c^
x	0.42	3.07	12.0	0.17	1.33	8.66	0.12	0.55	8.8
SD	0.38	3.78	4.70	0.17	1.45	7.01	0.09	0.71	5.99
Mode	0.25	1	16	0.125	1	16	0.0313	0.5	8
Median	0.25	2	16	0.125	1	8	0.06	0.5	8
MIC50	0.25	2	16	0.125	1	8	0.06	0.5	8
MIC90	0.5	8	16	0.5	2	16	0.25	2	16
ECV 95%	0.5	-	-	0.5	-	-	0.25	-	-
MIC > ECV	2/4.7	38/88.4	43/100	1/2.3	27/62.8	36/83.7	0	24/55.8	41/95.3
*Malassezia pachydermatis* CBS 1879									
Range	0.5	1	16 ^d^	0.125 ^e^	0.125–0.25 ^e,f^	8	0.125 ^e,g^	0.125 ^e,f,g^	16 ^d^
x	0.5	1	16	0.125	0.17	8	0.125	0.125	16
SD	0	0	0	0	0.07	0	0	0	0
Mode	0.5	1	16	0.125	0.125	8	0.125	0.125	16
Median	0.5	1	16	0.125	0.125	8	0.125	0.125	16
MIC > ECV	0	3/100	3/100	0	0	3/100	0	0	3/100

PC—planktonic cells, AC—adhered cells, MB—mature biofilm, x—average mean, SD—standard deviation, MIC—minimum inhibitory concentration, ECV—epidemiological cut-off value. ^a–g^—MIC values with the same designation are not statistically significantly different (*p* > 0.05).

**Table 4 jof-08-01209-t004:** Evaluation of susceptibility of planktonic cells (PC), adhered cells (AC) and mature biofilm (MB) of *M. pachydermatis* strains to tested antimycotics.

Susceptibility	Itraconazole			Voriconazole			Posaconazole		
	PC	AC	MB	PC	AC	MB	PC	AC	MB
S (*n*/%)	24/55.8	2/4.7	0	33/76.8	5/11.6	0	22/51.1	4/9.3	0
S-DD (*n*/%)	17/39.5	3/6.9	0	9/20.9	11/25.6	7/16.3	21/48.9	15/34.9	2/4.7
R (*n*/%)	2/4.7	38/88.4	43/100	1/2.3	27/62.8	36/83.7	0	24/55.8	41/95.3

S—susceptible, S-DD—susceptible dose-dependent, R—resistant, PC—planktonic cells, AC—adhered cells, MB—mature biofilm, *n*—number of isolates.

## Data Availability

Not applicable.

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
