# Peer review of "In Vitro Biofilm Formation by Malassezia pachydermatis Isolates and Its Susceptibility to Azole Antifungals"

_jof, 2022, doi:10.3390/jof8111209_

Round 1

Reviewer 1 Report

The authors study the ability to produce biofilms of M. pachydermatis strains isolated from the ear canal of dogs and their susceptibility to three azole.

Line 36: Delete comma in citations.

Line 67-68: Match font size

Row 185 to 190: major or minor inconsistency in symbols between text and table 4

Line 203: MB mature cell? In the text "MB"  it is mature biofilms. To clarify

Caption tab 3: explanations of the superscript letters are missing. Insert meaning of "mode"

Line 217: Close the parenthesis after the percentage

Discussion: Format the font size to the rest of the text

Line 230: replace the period with the comma after quotation 5.

Author Response

All suggested corrections were made directly in the text.

Reviewer 2 Report

Dear Authors,

The study entitled “In vitro biofilm formation by Malassezia pachydermatis iso- 2

lates and its susceptibility to azole antifungals” brings relevant information about the ability of Malassezia pachydermatis to produce biofilm and its susceptibility to azoles. The study is well written and design, but some comments are found below to make the manuscript suitable for publication.

1) Some information is missing in the Introduction to provide a better background of the fungal infection in dogs. What is the incidence of otitis caused by Malassezia in dogs? Is there any data regarding the epidemiology of these diseases? It is important to highlight the impact of this mycosis and the concerning aspect of reporting resistant strains.

2) Why the 9 strains that do not produce biofilm was removed from the study. It could be interest to compare the susceptibility of them (as planktonic cells) with those able to produce biofilms

3) Why did the authors use only azoles in the experiments? Especially considering that polyenes are also used to treat Malassezia infections in dogs, it could be interest to test it as well.

4) The authors mentioned some works in the Discussion that already studied the ability of M. pachydermatis to form biofilm and its susceptibility to antifungals. Considering that the authors bring similar data, it might be better to organize the Discussion highlighting what is already known in the literature and which role the study plays to improve the acknowledge of Malassezia infections in dogs. To present some different data, I would recommend exploring the biofilms in more details. For example, you could perform some electron microscopy to evaluate the structure of Malassezia biofilms, or even use some fluorescent probes to check the thickness of the extracellular matrix, which could also be performed by confocal microscopy. My major concern is about what new information the study brings compared to what is already known in the literature.

Author Response

Dear reviewer,

Thank you for all comments you made in the review.

In the text below, you will find our commentary on each paragraph of your reminders.

  1. Some information is missing in the Introduction to provide a better background of the fungal infection in dogs. What is the incidence of otitis caused by Malassezia in dogs? Is there any data regarding the epidemiology of these diseases? It is important to highlight the impact of this mycosis and the concerning aspect of reporting resistant strains.

In the Introduction, we included brief information on the incidence and epidemiology of Malassezia pachydermatis diseases in dogs.

2. Why the 9 strains that do not produce biofilm was removed from the study. It could be interest to compare the susceptibility of them (as planktonic cells) with those able to produce biofilms

We studied antifungal susceptibility only in biofilm forming strains. Strains that were not found to be biofilm producers were excluded from this experiment.

3. Why did the authors use only azoles in the experiments? Especially considering that polyenes are also used to treat Malassezia infections in dogs, it could be interest to test it as well.

We agree with the opinion of the opponent, but we selected only azole antifungals for this study.

4. The authors mentioned some works in the Discussion that already studied the ability of M. pachydermatis to form biofilm and its susceptibility to antifungals. Considering that the authors bring similar data, it might be better to organize the Discussion highlighting what is already known in the literature and which role the study plays to improve the acknowledge of Malassezia infections in dogs. To present some different data, I would recommend exploring the biofilms in more details. For example, you could perform some electron microscopy to evaluate the structure of Malassezia biofilms, or even use some fluorescent probes to check the thickness of the extracellular matrix, which could also be performed by confocal microscopy. My major concern is about what new information the study brings compared to what is already known in the literature.

Unfortunately, we did not use any mentioned method confirming the reduction of biofilm thickness after antifungal exposure.

The text in the discussion has been edited, we have tried to explain  the main meaning of this study more in detail.

Round 2

Reviewer 2 Report

Dear Authors,

The manuscript has been now improved and I believe it is suitable for publication.